# Why are undergraduate emerging adults anxious and avoidant in their romantic relationships? The role of family relationships

**Marta Díez**[☉], **Inmaculada Sánchez-Queija**[iD]*[☉], **Águeda Parra**[☉]

Dpto. Psicología Evolutiva y de la Educación (Developmental and Educational Psychology), Universidad de Sevilla, Sevilla, Spain

☉ These authors contributed equally to this work.
* queija@us.es

**Data Availability Statement:** All relevant data are within the manuscript and its Supporting Information files and at https://www.protocols.io/view/protocol-romantic-relationships-and-family-

## Abstract

The exploration of and search for romantic relationships is one of the developmental tasks that characterise emerging adulthood, a new developmental phase halfway between adolescence and full adulthood. This study aims to explore, in a Mediterranean country, the existing relationships between the subjective perception of some parental behaviour and the anxiety and avoidance dimensions of attachment during emerging adulthood. To do so, 1,502 university students (903 women and 599 men) aged between 18 and 29 (M = 20.32 and SD = 2.13) completed a self-report questionnaire. The results revealed that perceived family support and perceived parental warmth were negatively associated with the avoidance and anxiety dimensions. In contrast, perceived parental control (both behavioural and psychological) was found to be positively associated with both attachment dimensions. Perceived behavioural control was also found to play a moderator role between perceived parental warmth and romantic attachment anxiety. Only in cases in which emerging adults of our sample perceived low levels of behavioural control was warmth found to be negatively associated with anxiety. The main conclusion of this work is the negative impact that parental control seems to have on romantic attachment during emerging adulthood. The results are discussed with a focus on the continuing importance of the family context in relation to the completion of developmental tasks, even during emerging adulthood.

## Introduction

Emerging adulthood is a developmental stage characterised by individuals' search for identity, mainly through love and work. It is also a time of intense instability and great freedom, as well as a period full of possibilities in which young people feel themselves to be halfway between adolescence and adulthood [1, 2]. Emerging adults are no longer adolescents, but not quite yet adults either, and while they are less behaviourally and emotionally dependent on their parents, they are not (in the majority of cases, at least) completely autonomous individuals [1], and thus maintain their connectedness to their parents [3, 4]. Indeed, during emerging adulthood, the family

in-span-67dhhi6. doi: dx.doi.org/10.17504/
protocols.io.67dhhi6.

**Funding:** This study was supported by a grants
from the Ministerio de Economía y Competitividad
(Spanish Ministry of Economy and
Competitiveness, http://www.mineco.gob.es/
portal/site/mineco?lang_choosen=en) EDU2013-
45687-R to AP, the Ministerio de Ciencia,
Innovación y Universidades (Spanish Ministry of
Science, Innovation and Universities, http://www.
ciencia.gob.es/portal/site/MICINN/?lang_choosen=
en) RTI2018-097405-B-I00 to IS-Q. The funders
had no role in study design, data collection and
analysis, decision to publish, or preparation of the
manuscript.

**Competing interests:** The authors have declared
that no competing interests exist.

continues to be the main provider of care and security, as well as a central reference point for hopes and plans for the future [5, 6]. This key role is particularly relevant in southern European countries, which have often been linked to the "family welfare regime" [7], in reference to (among other issues) their traditions and strong family ties. Perhaps the clearest example of this family interdependence is the age at which young people leave home, with the mean age for this key landmark in a young adult's life being 29.3 years in Spain, according to Eurostat-2019 [8].

During emerging adulthood, young people begin to explore romantic relationships in more depth. However, in southern European countries, this exploration takes place under the umbrella of the nuclear family, with whom most young adults continue to live. It also coincides with their attempts to maintain a delicate balance between the search for intimacy and support in romantic relationships and the satisfaction of their own needs: studies, a job or search for a job and exploration of their own identity, etc. [9]. The exploration of and search for dyadic intimate relationships is one of the inherent developmental tasks of emerging adulthood [10], and during this phase, romantic relationships are much more oriented towards greater emotional closeness, stability and care for one's partner than during adolescence [11]. In this sense they are more similar to adult romantic relationships.

One of the main approaches to studying adult romantic relationships is attachment theory [12–14]. The first authors to study attachment in adults were Hazan and Shaver [15], who argued that the child attachment theory developed by Bowlby [16–18] could also be applied to romantic relationships. A short time later, Bartholomew [19] proposed an adult attachment model based on two dimensions: the image of the self and the image of the other. This model was empirically developed by Brennan et al. [20], who found that the image of the self was linked to anxiety in romantic relationships, whereas the image of the other was linked to avoidance. People with high attachment anxiety levels (with a negative image of themselves) tend to overvalue relationships, clinging to them and fearing separation. On the other hand, those who score highly for avoidance (with a negative image of others) minimise the importance of relationships and fear involvement and commitment [14, 21].

Beyond the field of attachment theory, few studies have analysed how family relationships are linked to the establishment of emotional bonds outside the family environment, and those that have were mainly conducted with children, focusing on how affection, warmth and family support foster good peer relations [22, 23]. One of the few studies carried out with emerging adults is that by Conger et al. [24], which analyses the influence of parenting styles on romantic relationships, finding that parenting styles characterised by high levels of warmth and parental monitoring and a lack of authoritarian discipline strongly predicted warmth and low hostility in romantic relationships. Other studies have also found that perceived parental support, warmth and sensitivity are associated with a greater degree of intimacy, more satisfactory relations and a more mature attitude towards romantic relationships [25–27]. Similarly, other authors have observed that adolescents and emerging adults who described their parents as more controlling and overprotective, with excessive use of punitive and intrusive control, had higher levels of attachment anxiety [28–30].

In sum, previous research has shown that young people's perceptions of good family support influence the establishment of healthy romantic relationships, both in adolescence [31] and in emerging adulthood [24, 27, 32]. Furthermore, parental warmth is also associated with positive romantic relationships during both adolescence [33–35] and emerging adulthood [25].

Nevertheless, results regarding the role of family control are less consistent. Thus, in a study carried out with young people aged between 14 and 21, Auslander et al. [36] found a positive association between parental control and satisfaction with romantic relationships, while in another study conducted with young people aged between 19 and 26, Karataç et al. [29] found that parental control was linked to greater anxiety in romantic relationships. This

contradiction may partly be due to the difference between behavioural and psychological control. It seems that when the control exercised by parents is psychological in nature, i.e., is a more intrusive type of control aimed at manipulating young people's thoughts and feelings [37], its influence on romantic relationships is negative during adolescence [38] and is maintained during emerging adulthood [29, 38, 39].

However, this association between psychological control by parents and romantic attachment has been called into question in Mediterranean or southern European cultures. For example, in a study with adolescents, Güngor and Bornstein [35] found that while greater psychological control by parents was linked to greater avoidance in the attachment relationship among the Belgian subsample, this association was not found in the Turkish subsample. These results confirm the need to analyse the role of family in the formation of romantic attachment outside northern European cultures.

Behavioural control, characterised by the establishment of limits and the supervision of children's behaviour [40], has been shown to be necessary and healthy during adolescence [34]. However, although few studies have focused on this particular issue, behavioural control seems to be a negative variable during emerging adulthood, since it hampers young people's development during this period [41, 42] and may also interfere with the establishment of healthy romantic relationships. Whatever the case, exploring the relationship between behavioural control during emerging adulthood and romantic attachment is particularly important due to the scarcity of extant research in this field.

Given that achieving a stable romantic relationship is a developmental task inherent to emerging adulthood [10], it is important to understand the factors which influence its attainment. In light of the above, the aim of this study was to explore, in Spain, the relationship between the anxiety and avoidance dimensions of romantic attachment during emerging adulthood and subjective perception of family support, parental warmth and behavioural and psychological control.

In line with the literature reviewed above, our initial hypotheses were that high perceived parental warmth and support would be positively linked to low attachment avoidance and anxiety in romantic relationships; we also expected perceived psychological control by parents to be linked to higher levels of attachment anxiety in romantic relationships. It is very difficult to predict the effect of psychological control on avoidance and of behavioural control on both dimensions of romantic attachment since, as explained above, existing evidence is too scarce to enable a robust hypothesis.

## Methods

### Participants

The sample comprised 1,502 university students (39.9% men, Mean age = 20.32, SD age = 2.13, Range age = 18–29, although 91.6% of the sample was between 18 and 23 years old. As regards perceived purchasing power, 15.8% claimed to have a low income, 70.1% a medium income and 14.1% a medium-high income level. The sample was collected from the University of Seville (US) and the University of Basque Country (UPV/EHU). Data were gathered from a representative number of students from different fields of study: 32.2% from social sciences and legal studies, 29.2% from health sciences, 23.5% from engineering and architecture, 8.5% from arts and humanities, and 6.7% from the sciences.

### Instruments

**Demographic variables.** All participants indicated their age, sex and perceived income level.

**Psychological variables.** Experience of close relationships: The Spanish version [43] of the questionnaire designed by Brenan et al. [20] was used for the data collection process. This questionnaire evaluates how an individual behaves in relationships with a partner across two dimensions: anxiety (e.g. I worry about being abandoned, $\alpha$ = .86) and avoidance (I prefer not to show a partner how I feel deep down, $\alpha$ = .84). They were rated on a 4-point scale ranging from 1 (completely disagree) to 4 (completely agree).

Social support: Was evaluated using the Multidimensional Scale of Perceived Social Support [44]. This scale comprises 4 items, with responses given on a Likert-type scale ranging from 1 = Very strongly disagree to 7 = Very strongly agree (e.g. I can talk about my problems with my family, $\alpha$ = .90). High scores indicate high family social support.

Parental warmth: The Parent Warmth Subscale (e.g. My parent accepts me and likes me as I am) of the Perception of Parents Scale (POPS), College Student Version [45, 46] was used to measure parental warmth. This questionnaire consists of 6 items rated on a 7-point Likert-type scale ranging from 1 (not at all true) to 7 (very true). High scores indicate high warmth. The results revealed an $\alpha$ = .82.

Psychological control: l was measured using the corresponding subscale of the Parental Styles Scale [47]. Emerging adults answered eight questions on a 6-point scale ranging from 1 (completely disagree) to 6 (completely agree). Sample question: "He/she makes me feel guilty when I don't do what he/she wants". The Cronbach's alpha for this subscale was = .87. High scores indicate high levels of perceived psychological control.

Behavioural control: Was measured using items adapted for Emerging Adults from Kerr and Stattin's Control Subscale [48]. Emerging adults answered five questions on a 6-point scale ranging from 1 (completely disagree) to 6 (completely agree). Sample question: "My father/mother tries to set rules about what I do in my spare time". The Cronbach's alpha for this subscale was = .77. High scores indicate high levels of perceived behavioural control.

## Procedure

The data collection process took place during the second semester of the 2014–2015 academic year. During the initial phase, faculties from different subject areas at the Universities of Sevilla (US) and the University of Basque Country (UPV/EHU) were contacted in order to request their consent and to arrange the gathering of the data during class time. Specially trained members of the research team collected the data from participating students during one hour of class time. All participants were informed in writing of the aim of the study and assurance was given that the survey was both anonymous and confidential. The inclusion criteria were to be in the classroom at the time of data collection and to be between 18 and 29 years old. All students participated voluntarily. The study was approved by the Coordinating Committee for the Ethics of Biomedical Research in Andalusia (Spain).

## Data analysis

Firstly, descriptive analyses were performed for the anxiety and avoidance variables. Analyses were performed separately for each sex through analysis of variance (ANOVA). Subsequently, a bivariate correlation analysis was conducted between the variables included in the study. The third step was to carry out regression equations with attachment anxiety and avoidance in romantic relationships as dependent variables, in order to determine which family variables best explained the DVs. Finally, moderation analyses were then performed. The SPSS V.24 statistical software package was used, along with the Newsom's Simple1 macro [49] for the moderation analysis, and Jose's ModGraph programme [50] for presenting the results in graph form.

## Results

The descriptive analyses (Table 1) and the analysis of variance performed revealed no sex-related differences in attachment anxiety–$F$ (1, 1266) = 2.84; $p$ = .092-. Nevertheless, male participants were found to be more avoidant in their romantic relationships than their female counterparts–$F$ (1, 1266) = 31.51; $p < .001$; $\eta2$ = .024-, although the effect size was small.

Next, romantic attachment was studied in terms of its association with the family variables under study. To this end, bivariate Pearson correlation analyses were conducted between the two dimensions, attachment anxiety and avoidance in romantic relationships, and the rest of the quantitative study variables (Table 2).

Although some correlations show low values, the results revealed that perceived social support and parental warmth were negatively and significantly correlated with attachment anxiety and avoidance in romantic relationships. Thus, those who scored higher for anxiety and avoidance also reported less family social support and a lower level of perceived parental warmth. Control (both behavioural and psychological) correlated positively and significantly with romantic attachment anxiety and avoidance, with greater parental control being linked to higher levels of anxiety and avoidance.

Regression equations (Table 3) were carried out using the insert method to determine which variables best explained attachment anxiety and avoidance in romantic relationships.

When all variables in the regression equation were controlled for, family social support failed to enter any of the regression equations. Warmth was associated with both anxiety and avoidance. In this last case, the relationship was negative, as hypothesised. However, when the rest of the study variables were controlled for, the relationship between warmth and anxiety was found to be positive, the opposite of both that hypothesised and that found in the correlation analyses (Table 2). Control (both behavioural and psychological) emerged as an important factor related to anxiety in romantic relationships, with young people who felt less controlled (behaviourally and psychologically) by their families scoring lower in this dimension. Finally, behavioural control was also positively related to attachment avoidance in romantic relationships, but the association between avoidance and psychological control was not significant. The effect size of the model was low ($R^2$ = .09 and .04). However, it provides interesting data on the relationship between the different family variables and attachment anxiety and avoidance in romantic relationships, even during emerging adulthood.

As mentioned earlier, the relationship between perceived parental warmth and anxiety changed from negative to positive between the correlations table (Table 2) and the regression equation (Table 3). In order to find an explanation for this change and bearing in mind the negative effect of control on development during emerging adulthood, we analysed the possible moderator effects of psychological and behavioural control on the relationship between parental warmth and both anxiety and avoidance.

The results for perceived behavioural control revealed a significant interaction ($p$ = .03) between warmth and romantic attachment anxiety. Specifically, the slope between warmth and anxiety was not significant when behavioural control was either medium ($p$ = .13) or high ($p$ = .87), although it was significant ($p$ = .028) when behavioural control was low. This finding,

**Table 1. Descriptive analyses.**

|  | Male | | | Female | | | Total | | |
|---|---|---|---|---|---|---|---|---|---|
|  | M | SD | Min-Max | M | SD | Min-Max | M | SD | Min-Max |
| Anxiety | 2.23 | .54 | 1.00–3.93 | 2.28 | .58 | 1.00–3.93 | 2.26 | .56 | 1.00–3.93 |
| Avoidance | 1.79 | .45 | 1.00–3.59 | 1.64 | .43 | 1.00–3.24 | 1.70 | .44 | 1.00–3.59 |

**Table 2. Bivariate correlation analysis between anxiety, avoidance and family variables.**

| | Anxiety ($r^2$) | Avoidance ($r^2$) |
|---|---|---|
| Family Social Support | -.09** | -.14** |
| Parental Warmth | -.09* | -.19** |
| Parental Behavioural Control | .26** | .11** |
| Parental Psychological Control | .26** | .11** |

*$p \leq .01$.

**$p \leq .001$.

**Table 3. Regression analyses on anxiety and avoidance.**

| | Anxiety (Beta) $R^2 = .09$ | Avoidance (Beta) $R^2 = .04$ |
|---|---|---|
| Family Social Support | -.05 | -.02 |
| Warmth | .09* | -.18*** |
| Behavioural Control | .16*** | .09** |
| Psychological Control | .20*** | -.04 |

***$p \leq .001$;

** $p \leq .01$;

* $p \leq .05$.

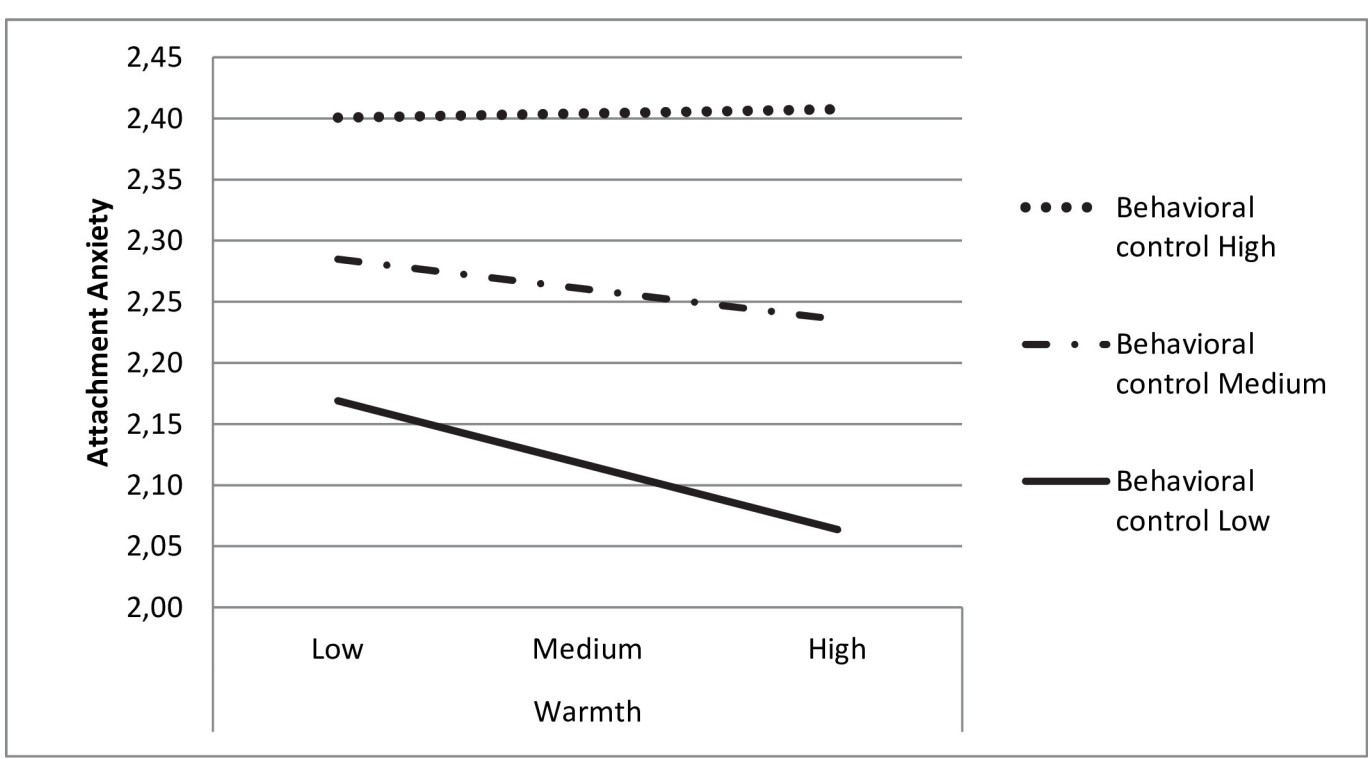

**Fig 1. Moderation of the effect of warmth on attachment anxiety by behavioural control.**

which is presented in Fig 1, indicates that only in cases in which the level of perceived behavioural control exerted by parents was low did the warmth they showed to their children help reduce the attachment anxiety that the undergraduate emerging adults in our study felt when establishing romantic relationships. Nevertheless, when these emerging adults perceived their parents as controlling, perceived warmth did nothing to reduce attachment anxiety.

Although the moderating effect of perceived behavioural control did not reach statistical significance ($p = .06$) in relation to perceived parental warmth and romantic attachment avoidance, the results did reveal that the lower the level of behavioural control, the stronger the negative relationship between warmth and avoidance ($\beta = .23$, $p < .001$; $\beta = .18$, $p < .001$; $\beta = .14$ $p < .001$, respectively). Finally, the moderator effect of psychological control on the relationship between warmth and romantic attachment anxiety in romantic relationships was not significant ($p = .14$).

## Discussion

The aim of this study was to determine the associations which exist between family variables and the anxiety and avoidance dimensions of romantic attachment during emerging adulthood. In line with our initial hypotheses, the results reveal that both perceived family support and perceived parental warmth are negatively associated with the two dimensions. Also, parental control (both behavioural and psychological) was found to positively correlate with attachment anxiety and avoidance in romantic relationships.

In addition to these results, in our opinion, the most important finding is the unexpected moderating role played by perceived behavioural control on the relationship between perceived parental warmth and attachment anxiety in romantic relationships. Thus, only when behavioural control is low is there an association between parental warmth and attachment anxiety, meaning that only when behavioural control is low does high parental warmth correlate with low attachment anxiety in romantic relationships. The results found in relation to avoidance were consistent with those observed in relation to anxiety, although, in this case, without reaching statistical significance: the lower the behavioural control, the stronger the negative association between perceived parental warmth and attachment avoidance. In other words, when behavioural control is perceived as low, greater warmth is more closely linked to lower levels of avoidance.

The descriptive results of this study revealed that male participants were more avoidant than their female counterparts. The fact that men were found to be more avoidant than women is consistent with that reported by other studies described in the meta-analysis on sex differences in romantic attachment carried out by Del Giudice [51]. This work also found that women scored higher than men for anxiety in all regions (including Europe) except Asia. In our study, however, and in line with the results reported by Karataç et al. [29] with Turkish emerging adults, no sex differences were observed in relation to attachment anxiety. This may be due to the fact that the emerging adult women in our sample—all of whom were undergraduate university students—were more self-confident than their counterparts in other segments of the female population, or at least as self-confident as their male classmates, a circumstance which (among other things) prevents them from feeling anxious in their romantic relationships. Alternatively, as Karataç et al. [29] argued, this could also be due to cultural differences. The high value attached to the traditional and patriarchal family in both Spanish and Turkish cultures [52] may give rise to a tendency to overvalue relationships, prompting emerging adults to cling to them and fear separation, thereby generating similar levels of anxiety in both sexes. Other studies carried out with Spanish university students failed to find any sex differences in any attachment style [53]. Nevertheless, this finding highlights the need to continue

analysing sex differences in the various dimensions of romantic relationship attachment during emerging adulthood in different cultures.

The findings reported in this study reveal that emerging adults with higher levels of attachment anxiety and avoidance in their romantic relationships also perceive lower levels of social support and parental warmth. In other words, poorer quality family relationships (in terms of affection and warmth) are associated with insecure romantic attachment. This result is consistent with those reported by other studies, which found that nurturant-involved parenting predicted romantic relationships based on warmth, support and low hostility [24, 33], which validates and gives consistency to the results found in this work, in spite of the low explanatory power of the model.

Avoidance and, above all, attachment anxiety in romantic relationships are also linked to perceived family control. Indeed, the highest Beta values in the regression equation that predicted anxiety were obtained for behavioural and psychological control, and other studies have demonstrated the negative association which exists between parental control and healthy romantic relationships [29, 33, 36], as well as the link between control and conflict in these same relationships [38]. Controlling parents who continue to supervise how their adult children spend their money or with whom they go out (behavioural control), or who use emotional blackmail to ensure that they continue to abide by their rules (psychological control), could cause their children to develop a negative self-image, conveying the idea that they cannot trust themselves and are incapable of taking charge of their lives, thereby fostering anxiety in their interpersonal relationships. This study, therefore, confirms that, in addition to psychological control, behavioural control by families is also related to unhealthy romantic relationships during emerging adulthood. These results highlight, once again, the negative role played by parental control (particularly behavioural control, for which less evidence exists) during emerging adulthood [40, 41].

Nevertheless, our results go one step further, since they shed light on the moderating role played by control in the relationship between warmth and attachment anxiety and avoidance in romantic relationships. Our results reveal that behavioural control moderates the relationship between parental warmth and anxiety, meaning that, when control is medium and high, warmth is no longer significantly associated with attachment anxiety. In other words, the negative effect of family behavioural control on romantic relationships is so strong that only when this control is low does parental warmth act as a protective factor against romantic relationship anxiety.

Probably, the behavioural control that was necessary in previous stages, in which boys and girls still needed to internalise external rules, has negative consequences in emerging adulthood. Emerging young adults—and, particularly, the university students in the sample—need autonomy, since they are already adults (and of legal age). Parents who still control their adult children are contributing to an internal working model promoting distrust in themselves and insecurity an insecurity that they will pass on to other attachment figures, such as their partners, in the form of avoidance (related to the model of oneself).

As regards avoidance, although the moderator effect is not significant, our results also indicate that warmth has a greater protective effect against romantic relationship avoidance when behavioural control is low.

Additionally, in connection with avoidance, it is important to note that, although psychological control was positively linked to romantic relationship avoidance, this type of control loses its explanatory power when the other family variables are included in the regression equation. In our opinion, this result is also worth highlighting, since it suggests that, although parental control is the most important family variable for understanding anxiety in romantic relationships during emerging adulthood, warmth is key to explaining avoidance in these same relationships. Receiving warmth and affection from one's family, even during the third

decade of life, fosters a favourable model of the other, with no fear of engagement and commitment in romantic relationships.

In sum, among the undergraduate emerging adults in our study, the situations most conducive to promoting healthy attachment in romantic relationships are those characterised by high perceived levels of family social support and warmth, coupled with low levels of both psychological and behavioural control.

The study has certain limitations. The most important limitation is related to the low explanatory power of the models and certain correlations. Evidently, there are explanatory variables of romantic attachment beyond those relating to parental behaviour that have not been considered in this research. However, family relationships also affect romantic attachment, and this work has not only indicated the direction of this relationship at a stage that has seldom been studied but, moreover, it has made progress addressing how different dimensions of the family system relate to the romantic attachment model. Secondly, having retrospective measures on perceived parent-child relationships during childhood and adolescence would have enabled us to explore the evolution of attachment, warmth and family control across different developmental stages, and would have enriched the analyses of the attachment anxiety and avoidance dimensions in romantic relationships during emerging adulthood. Thirdly, being a cross-sectional study, the results offer no information about the development and changes experienced by young people over time. Finally, the sample was made up exclusively of university students, we must be cautious when generalising our findings to older emerging adults and those who are not studying at a university.

Despite these limitations, this study, which was conducted with a broad sample of 1,500 participants, is a pioneering study in Spain and provides greater insight into how different family environment variables contribute to affecting romantic attachment models. The results indicate the negative role played by family control on the lives of young adults, since although this control is necessary during childhood and adolescence, it is superfluous and indeed harmful during this emerging adulthood. The study also offers some inspiring data, such as the absence of sex differences in relation to attachment anxiety. This finding indicates the need to continue analysing sex differences in the construction of adult attachment models and their possible determinants or correlates. This piece of research therefore adds to our knowledge of how to help young people to establish healthy romantic relationships, one of the principal developmental tasks of emerging adulthood, and does so by analysing the role played by family in this undertaking. Indeed, family continues to be a key development context for young people and influences their choice of romantic partner [54]. There are many young people who, in both Spain and many other Western countries, continue to live in the family home until well into their third decade of life. This is a new phenomenon that previous generations did not experience, and further research is required to enable a more thorough understanding of this period. In that sense, further research should continue to analyse how family dynamics, and especially parental control, relate to personal and relational adjustment during emerging adulthood. Only in this way will we be able to support positive parenting during emerging adulthood, thereby contributing to the psychological wellbeing of young people, including positive romantic relationships.

## Supporting information

**S1 Dataset. Datos romantic attachment.**
(SAV)

## Acknowledgments

We wish to acknowledge the statistical support provided by Dr. Carlos Camacho Martínez Vara de Rey.

## Author Contributions

**Conceptualization:** Marta Díez, Inmaculada Sánchez-Queija, Águeda Parra.

**Formal analysis:** Marta Díez, Inmaculada Sánchez-Queija.

**Funding acquisition:** Inmaculada Sánchez-Queija, Águeda Parra.

**Investigation:** Marta Díez, Inmaculada Sánchez-Queija, Águeda Parra.

**Methodology:** Inmaculada Sánchez-Queija.

**Resources:** Marta Díez, Inmaculada Sánchez-Queija, Águeda Parra.

**Supervision:** Inmaculada Sánchez-Queija, Águeda Parra.

**Writing – original draft:** Marta Díez, Inmaculada Sánchez-Queija.

**Writing – review & editing:** Marta Díez, Inmaculada Sánchez-Queija, Águeda Parra.

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
