## [Decision Letter · Decision Letter 0]

24 Jul 2019

PONE-D-19-17219

WHY ARE EMERGING ADULTS ANXIOUS AND AVOIDANT IN THEIR ROMANTIC RELATIONSHIPS? THE ROLE OF FAMILY RELATIONSHIPS

PLOS ONE

Dear Authors,

Thank you for submitting your manuscript to PLOS ONE. After careful consideration, we feel that it has merit but does not fully meet PLOS ONE’s publication criteria as it currently stands. Therefore, we invite you to submit a revised version of the manuscript that addresses the points raised during the review process.

We would appreciate receiving your revised manuscript by 08/23/19. To enhance the reproducibility of your results, we recommend that if applicable you deposit your laboratory protocols in protocols.io, where a protocol can be assigned its own identifier (DOI) such that it can be cited independently in the future. For instructions see: http://journals.plos.org/plosone/s/submission-guidelines#loc-laboratory-protocols

We look forward to receiving your revised manuscript.

Kind regards,

Luca Cerniglia, PhD

Academic Editor

PLOS ONE

Journal Requirements:

2. Please update the ethics statement in the Methods section (and Procedure section) of your manuscript file to remove the word "Blinded" and replace with the named IRB that approved your study.

Reviewers' comments:

Reviewer's Responses to Questions

**Comments to the Author**

1. Is the manuscript technically sound, and do the data support the conclusions?

Reviewer #1: Partly

Reviewer #2: Yes

2. Has the statistical analysis been performed appropriately and rigorously? 

Reviewer #1: Yes

Reviewer #2: Yes

3. Have the authors made all data underlying the findings in their manuscript fully available?

Reviewer #1: No

Reviewer #2: Yes

4. Is the manuscript presented in an intelligible fashion and written in standard English?

Reviewer #1: No

Reviewer #2: Yes

5. Review Comments to the Author

Reviewer #1: Thank you for the opportunity to review the manuscript titled “Why are emerging adults anxious and avoidant in their romantic relationships? The role of family relationships”.

This study aimed to explore the relationships which exist between family functioning and the anxiety and avoidance dimensions of attachment during emerging adulthood

The results are interesting but I think that, in order to be able to publish them, several revisions are necessary. Furthermore, I strongly recommend a proofreading from a native speaker and an academic service for the global structure of the manuscript. Please find below some comments.

Major revisions

In different parts of the text, confusion is made between several terms: “attachment models” “attachment styles” and “behaviors in relationships” or “romantic attachment”. These terms refer to very different constructs and it is important to clarify what is referred to in this study. What has been evaluated? The questionnaire Experience of close relationships does not assess individual attachment models attachment and it is important to clarify it in the text.

Moreover, authors stated that the study “aim to explore the relationships which exist between family functioning…”. However, family functioning has not been evaluated. What has been evaluated is the subjective perception of some parental behaviors.

Procedure

As regard to the sample of the study, it is composed of university students who have attended the second semester of lessons. Although their age coincides with that of "emerging adults", it should be specified (also in the title) that the sample is composed of university students and therefore is not so generalizable to all emerging adults.

Has written informed consent been obtained?

Results

Does Table 2 contain the Pearson coefficients?

If so, the values are extremely low. It is possible that the correlation is significant (given the sample size), but it must be considered that these coefficients have extremely low values.

In the Table 3 it would be important to enter the values of Adjusted R. Again, it should be noted that the R values are very low.

With respect to moderation analyses, it would be necessary to insert the values of the analyses, first of all the value of R change. Furthermore it would be necessary to insert the values of Δ.

Limitations

it is important to underline the low values of these analyses in the limitations. As highlighted by the authors, they may depend on the fact that they are retrospective data, but also that there are variables that have not been considered. Moreover, it is important to underline that these results are not to be generalized to all emerging adults, but concern a sample of university students.

Reviewer #2: Dear Editor, thank you for the opportunity of revising this manuscript titled “Why are emerging adults anxious and avoidant in their romantic relationships? The role of family relationships”. The study took in consideration, in a large sample of university students, the possible relationships between perceived family functioning and the anxiety and avoidance dimensions of romantic attachment during emerging adulthood. I read the article with interest and I believe that this is an interesting manuscript, adding scientific knowledge to previous literature and potentially impacting research in the field. The study is clear and well written. Introduction and Discussion section are comprehensive and effectively guide the reader through the topic.

I suggest the authors to address some minor points:

Abstract;

I suggest to insert a brief conclusion section in the abstract.

Introduction:

The study is specifically focused on the emerging adulthood, considering the role played by family functioning in the anxiety and avoidance dimensions of young adults’ romantic attachment.

The authors have well explained that emerging adulthood is a developmental stage with evolutionary tasks and phase-specific characteristics different from both adolescence and adulthood.

Moreover, the authors have reviewed very clearly the results of previous literature which have pointed out that the family dimensions being studied (i.e., parental warmth, support and family control family control) have a different effect on the individual in adolescence compared to young adults. So, given that the study took in consideration a sample of youths between 18 and 29 years old, I think it would be helpful to clarify the theoretical model on which the authors are based the definition of “emerging adulthood” respect to late adolescence stage.

This is not a longitudinal study, so authors should clarify already in the introduction because, and on the basis of which literature, it is possible to draw cause-effect conclusions in retrospective or cross-sectional studies.

Methods

What were the criteria for inclusion and exclusion of subjects?

The psychometric properties of the original version of the instruments and of the Spanish version have been reported for all instruments.

I suggest to better described the instrument Experience of close relationships. On the base of what criterion subjects obtain high or low scores of anxiety/avoidance? Is there a Likert scale? There is a cut-off?

In the subsection “Data analyses” it’s important to specify more precisely the type of analysis carried out. For example, the results show that an analysis of the difference between the averages of males and females was carried out. Was an ANOVA conducted? This should be specified (also in the result section). Similarly, are the correlation analyses conducted Pearson correlational analyses?

The age of the sample varies from 18 to 29 years of age. As the age range is very wide, has the possible effect of age been considered?

Results and discussion

At line 215 authors said that “romantic partner attachment was studied in terms of its association with

the variables under study”. I suggest that you refer to “family variable” under study

In the regression model, have you considered the possible role played by gender? I think this aspect is important, given that men were found to be more avoidant than women

One of the main interesting results was that parental behavioral control moderates the relationship between parental warmth and anxiety. The authors should enrich the discussion of this aspect, providing a theoretical hypothesis underlying this interesting finding. It would also be useful to broaden the conclusions, with the implications for further research.

6. PLOS authors have the option to publish the peer review history of their article (what does this mean?). If published, this will include your full peer review and any attached files.

Reviewer #1: No

Reviewer #2: No

---

## [Author Response · Author response to Decision Letter 0]

4 Oct 2019

We are very thankful for all the comments and suggestions made by the reviewers concerning our manuscript entitled Why are undergraduate emerging adults anxious and avoidant in their romantic relationships? The role of family relationships. All of the comments and suggestions have undoubtedly contributed to the improvement of its quality and adjustment to the standards of PLOS-One. We particularly appreciate all comments that enhance the consistency of the manuscript.

We will now proceed to outline all the changes that have been made in the manuscript in order to meet the suggestions of the reviewers.

Comments to the Author

Reviewer #1: Thank you for the opportunity to review the manuscript titled “Why are emerging adults anxious and avoidant in their romantic relationships? The role of family relationships”.

This study aimed to explore the relationships which exist between family functioning and the anxiety and avoidance dimensions of attachment during emerging adulthood.

The results are interesting but I think that, in order to be able to publish them, several revisions are necessary. Furthermore, I strongly recommend a proofreading from a native speaker and an academic service for the global structure of the manuscript. Please find below some comments.

Major revisions

In different parts of the text, confusion is made between several terms: “attachment models” “attachment styles” and “behaviors in relationships” or “romantic attachment”. These terms refer to very different constructs and it is important to clarify what is referred to in this study. What has been evaluated? The questionnaire Experience of close relationships does not assess individual attachment models attachment and it is important to clarify it in the text.

We really appreciate this comment. We agree with the reviewer and have changed different expressions throughout the manuscript to: Romantic attachment and two dimensions: romantic attachment anxiety and romantic attachment avoidance. Moreover, in order to be much more consistent, we have changed other expressions such as “family affect” (now referred to as “perceived parental warmth” which is more accurate)

Moreover, authors stated that the study “aim to explore the relationships which exist between family functioning…”. However, family functioning has not been evaluated. What has been evaluated is the subjective perception of some parental behaviors. 

Again, we agree with the reviewer. Expression has been changed.

Procedure

As regard to the sample of the study, it is composed of university students who have attended the second semester of lessons. Although their age coincides with that of "emerging adults", it should be specified (also in the title) that the sample is composed of university students and therefore is not so generalizable to all emerging adults.

This information was already included in the abstract section: 1,502 university students. However, we have added “undergraduate” in the title and throughout the article, and we included the fact as a limitation in the discussion section.

Has written informed consent been obtained?

The conditions of anonymity and the confidentiality policy of the data were written in the header of the questionnaires and were explained before handing out the booklets in the classroom. In this way, only those persons who agreed with these conditions filled in the questionnaires and participated in the research. When the students filled in the questionnaires, they consented to participate in the research. We believe that this information is clearly stated in the procedure and, in any case, the “in writing” statement has been added to clarify it:

“All participants were informed in writing of the aim of the study and assurances were given that the survey was both anonymous and confidential. All students participated voluntarily”.

Results

Does Table 2 contain the Pearson coefficients?

If so, the values are extremely low. It is possible that the correlation is significant (given the sample size), but it must be considered that these coefficients have extremely low values. 

It has been specified that bivariate correlations had been calculated by Pearson correlations and added in the text effect size of correlations.

In the Table 3 it would be important to enter the values of Adjusted R. Again, it should be noted that the R values are very low.

This information has been added: R 2 and low effect size.

With respect to moderation analyses, it would be necessary to insert the values of the analyses, first of all the value of R change. Furthermore it would be necessary to insert the values of Δ.

R2 values of total regression model of interaction have been added. However, simple1 macro uses the introduction method to conduct the regression analysis, and that is because R change or Δ is not provided. However, simple1 brings a p value for the slopes, a very valuable indicator that is easy and intuitive to interpret. That is why we have chosen it.

Limitations

it is important to underline the low values of these analyses in the limitations. As highlighted by the authors, they may depend on the fact that they are retrospective data, but also that there are variables that have not been considered. Moreover, it is important to underline that these results are not to be generalized to all emerging adults, but concern a sample of university students. 

Once again, we agree with the reviewer and this information has been added twice in the discussion section.

Reviewer #2: Dear Editor, thank you for the opportunity of revising this manuscript titled “Why are emerging adults anxious and avoidant in their romantic relationships? The role of family relationships”. The study took in consideration, in a large sample of university students, the possible relationships between perceived family functioning and the anxiety and avoidance dimensions of romantic attachment during emerging adulthood. I read the article with interest and I believe that this is an interesting manuscript, adding scientific knowledge to previous literature and potentially impacting research in the field. The study is clear and well written. Introduction and Discussion section are comprehensive and effectively guide the reader through the topic.

I suggest the authors to address some minor points:

Abstract;

I suggest to insert a brief conclusion section in the abstract. 

Thank you. This has been done.

Introduction:

The study is specifically focused on the emerging adulthood, considering the role played by family functioning in the anxiety and avoidance dimensions of young adults’ romantic attachment.

The authors have well explained that emerging adulthood is a developmental stage with evolutionary tasks and phase-specific characteristics different from both adolescence and adulthood.

Moreover, the authors have reviewed very clearly the results of previous literature which have pointed out that the family dimensions being studied (i.e., parental warmth, support and family control family control) have a different effect on the individual in adolescence compared to young adults. So, given that the study took in consideration a sample of youths between 18 and 29 years old, I think it would be helpful to clarify the theoretical model on which the authors are based the definition of “emerging adulthood” respect to late adolescence stage. 

We agree on the theoretical interest in differentiating between late adolescence and emerging adulthood. However, since the article focuses on the existing relationship between parental behaviours and the dimensions of anxiety and avoidance of romantic attachment, we believe that it would not be appropriate to discuss whether the concept of emerging adulthood is better or worse than that of late adolescence. We prefer it if we consider that there is a clear qualitative cut at 18, marked by the established legal age. Reaching this legal age requires the assumption of responsibilities, or at least social responsibilities.

This is not a longitudinal study, so authors should clarify already in the introduction because, and on the basis of which literature, it is possible to draw cause-effect conclusions in retrospective or cross-sectional studies.

We really appreciate this comment due to the fact that it improves the consistency of the manuscript. Throughout the manuscript, we have avoided alluding to relations of causality, speaking of association or effect, but not about cause-effect. Cause-effect relationships require that facts be related or proximate in time, or that one precedes the other. If this type of relationship can be inferred at any time, it is because we have considered that family relationships precede couple relationships. Even so, we have reviewed the text and paid special attention so as not to allude to causality. 

Methods

What were the criteria for inclusion and exclusion of subjects?

This information has been included in the procedure section.

The psychometric properties of the original version of the instruments and of the Spanish version have been reported for all instruments.

I suggest to better described the instrument Experience of close relationships. On the base of what criterion subjects obtain high or low scores of anxiety/avoidance? Is there a Likert scale? There is a cut-off? 

ECR has no cut-off points. It is a Likert scale. It has been added to the text. 

In the subsection “Data analyses” it’s important to specify more precisely the type of analysis carried out. For example, the results show that an analysis of the difference between the averages of males and females was carried out. Was an ANOVA conducted? This should be specified (also in the result section). Similarly, are the correlation analyses conducted Pearson correlational analyses? 

We agree with the reviewer and this has been done.

The age of the sample varies from 18 to 29 years of age. As the age range is very wide, has the possible effect of age been considered? 

We agree with the reviewer that 18 year-old and 29 year-old participants could be very different. However, as is described in the participant section, the sample has an average of 20.32 years old, a SD of 2.13, and all are undergraduate students. Moreover, the majority of the sample was under 23 years old. That is because age has no relationship with the variables under study and has not been considered. We have added this information to clarify:

 Mean age= 20.32, SD age= 2.13, Range age= 18-29, although 91.6% of the sample was between 18 and 23 years old).

Results and discussion

At line 215 authors said that “romantic partner attachment was studied in terms of its association with the variables under study”. I suggest that you refer to “family variable” under study.

Thank you. This has been done.

In the regression model, have you considered the possible role played by gender? I think this aspect is important, given that men were found to be more avoidant than women. 

As reviewer 1 points out, the model has a low R2. Our main interest in this paper was to understand the relationships between family variables and romantic attachment dimensions, so we have selected these variables. In any case, we have explored sex interaction and they were not significant.

One of the main interesting results was that parental behavioral control moderates the relationship between parental warmth and anxiety. The authors should enrich the discussion of this aspect, providing a theoretical hypothesis underlying this interesting finding. It would also be useful to broaden the conclusions, with the implications for further research. 

Possible implications for further research have been added to the discussion.

---

## [Editor Report · Decision Letter 1]

8 Oct 2019

Why are undergraduate emerging adults anxious and avoidant in their romantic relationships? The role of family relationships

PONE-D-19-17219R1

Dear Authors,

We are pleased to inform you that your manuscript has been judged scientifically suitable for publication and will be formally accepted for publication once it complies with all outstanding technical requirements.

With kind regards,

Luca Cerniglia, PhD

Academic Editor

PLOS ONE
---

## [Editor Report · Acceptance letter]

22 Oct 2019

PONE-D-19-17219R1 

Why are undergraduate emerging adults anxious and avoidant in their romantic relationships? The role of family relationships 

Dear Dr. Sánchez-Queija:

I am pleased to inform you that your manuscript has been deemed suitable for publication in PLOS ONE. Congratulations! Your manuscript is now with our production department. 

With kind regards,

on behalf of

Dr. Luca Cerniglia 

Academic Editor

PLOS ONE